# Induction of Stem-Cell-Derived Cardiomyogenesis by Fibroblast Growth Factor 10 (FGF10) and Its Interplay with Cardiotrophin-1 (CT-1)

**DOI:** 10.3390/biology11040534

**Published:** 2022-03-30

**Authors:** Farhad Khosravi, Negah Ahmadvand, Maria Wartenberg, Heinrich Sauer

**Affiliations:** 1Department of Physiology, Justus-Liebig University Giessen, Aulweg 129, 35392 Giessen, Germany; heinrich.sauer@physiologie.med.uni-giessen.de; 2Cardio-Pulmonary Institute, Universities of Giessen and Marburg Lung Center (UGMLC), German Center for Lung Research (DZL), Justus-Liebig University Giessen, Aulweg 130, 35392 Giessen, Germany; negah.ahmadvand@innere.med.uni-giessen.de; 3Department of Internal Medicine I, Division of Cardiology, University Hospital Jena, Friedrich Schiller University, 07743 Jena, Germany; maria.wartenberg@med.uni-jena.de

**Keywords:** stem cell therapies, FGF10 (fibroblast growth factor 10), cardiotrophin-1 (CT-1), heart regeneration, cardiomyocyte proliferation, cardiomyogenesis

## Abstract

**Simple Summary:**

Cardiovascular diseases are worldwide one of the leading contributors of mortality, and among multiple therapeutic approaches, stem cell therapy has been introduced as a robust therapeutic strategy to alleviate related symptoms and restore cardiac functions. Prior to this, however, for successful cell therapy, an adequate number of functional and safe cardiac cells needs to be generated. For this purpose, our approach was boosting the proliferative capacity of stem cell-derived cardiomyocytes using growth factors, such as fibroblast growth factor 10 (FGF10) and cardiotrophin-1 (CT-1). Our results demonstrated that FGF10 and CT-1 substantially increased the number of cardiac cells that originated from stem cells. In molecular assays, to assess RNA and protein level alterations, the enhanced presence of specific markers for cardiac cells after treatment of stem cells with FGF10 and/or CT-1 was confirmed. This inducing potential of cardiac cells can particularly be applicable in the cell replacement-based therapies of cardiac infarction. This research sheds light on the putative effect of FGF10 and CT-1 in the transition of stem cells to cardiac cells, leading to the repair and survival of the heart.

**Abstract:**

For heart regeneration purposes, embryonic stem cell (ES)-based strategies have been developed to induce the proliferation of cardiac progenitor cells towards cardiomyocytes. Fibroblast growth factor 10 (FGF10) contributes to cardiac development and induces cardiomyocyte differentiation in vitro. Yet, among pro-cardiogenic factors, including cardiotrophin-1 (CT-1), the hyperplastic function of FGF10 in cardiomyocyte turnover remains to be further characterized. We investigated the proliferative effects of FGF10 on ES-derived cardiac progenitor cells in the intermediate developmental stage and examined the putative interplay between FGF10 and CT-1 in cardiomyocyte proliferation. Mouse ES cells were treated with FGF10 and/or CT-1. Differential expression of cardiomyocyte-specific gene markers was analyzed at transcript and protein levels. Substantial upregulation of sarcomeric α-actinin was detected by qPCR, flow cytometry, Western blot and immunocytochemistry. FGF10 enhanced the expression of other structural proteins (MLC-2a, MLC-2v and TNNT2), transcriptional factors (NKX2-5 and GATA4), and proliferation markers (Aurora B and YAP-1). FGF10/CT-1 co-administration led to an upregulation of proliferation markers, suggesting the synergistic potential of FGF10 + CT-1 on cardiomyogenesis. In summary, we provided evidence that FGF10 and CT-1 induce cardiomyocyte structural proteins, associated transcription factors, and cardiac cell proliferation, which could be applicable in therapies to replenish damaged cardiomyocytes.

## 1. Introduction

Strategies to restore cardiac physiological functions through the induction of cardiomyocyte proliferation aim to address the limited regenerating capacity of cardiac cells after birth. The boosted repairing potential of stem cells may alleviate the outrageous mortality and morbidity of cardiac diseases accompanied by the industrialized way of life [1,2]. FGF10 belongs to the FGF superfamily comprising 22 identified members. These FGF ligands with multi-functional roles in the development, homeostasis and pathology of the heart have received special research interests in recent years for the induction of cardiomyogenesis. FGF ligands mediate their actions in the heart through interactions with co-receptors (heparin/heparan sulfates or Klothos) and FGF receptors (FGFRs 1b, 1c, 2b, 2c, 3b, 3c and 4) [2,3]. In the early stages of cardiac development, FGF10 engages heparan sulfate and FGFR2b [4]. FGFR2b expression and subsequent FGF10-mediated effects occur in cardiomyocytes and have not been detected in cardiac fibroblasts [5]. Regarding the differential expression of FGFs in distinct regions in early heart development, FGF10 is expressed in the pharyngeal mesoderm. As a result, this ligand is considered a specific marker of the second heart field (SHF) [6]. FGF10, together with FGF8, contributes to the proliferation of progenitor cells of the SHF and, consequently, arterial pole development [7]. In the embryonic stage, a lethal homozygote ablation of Fgf10 impairs cardiac formation along with ventricular transposition [4,5]. Nonetheless, proper elongation of the arterial pole and septation in the absence of FGF10 may denote that FGF10 function is negligible in arterial development. However, severe cardiac impairments in SHF deployment coincide with the deletion of the predominant receptor of FGF10 (i.e., FGFR2b) [4]. Since FGF8 is another ligand for the FGF10 receptor, and the lack of both *Fgf8* and *Fgf10* exacerbates OFT and right ventricle dysmorphogenesis, a functional redundancy may occur between FGFs 8 and 10 in early heart development [4,7]. The function of FGF10 could overlap with FGF3, which is essential for the coordination of cardiac progenitor cells once the heart tube elongates [8].

In the developmental stage of the human embryo, FGF10 expression occurs in the outflow tract (OFT), where FGF10 functions in the differentiation of cardiomyocytes from embryonic stem cells. In congenital heart disease patients, mutations in FGF10 (together with FGF8) give rise to OFT impairment and consequently conotruncal defects, enhancing the risk of mortality [9].

FGF10 is among an array of ligands in this family (including FGFs 1, 2, 8 and 16), playing a role in the proliferation of cardiomyocytes [7,10,11,12]. Particularly in the context of regeneration (e.g., post-myocardial infarction therapies), this potential could serve the induction of proliferation in cardiomyocytes to counteract the dwindled proliferative capacity occurring during embryogenesis and exacerbating postnatally [13,14,15]. FGF10’s contribution to cardiomyocyte proliferation was further confirmed after observing ventricular alterations, particularly in the phenotype of the right ventricle in mutant models. Mechanistically, the regulatory role of FGF10 in the proliferation of cardiomyocytes is associated with the downregulation of cyclin-dependent kinase inhibitor p27^kip1^ as a consequence of phosphorylation of the forkhead box O3 (FOXO3) transcription factor [5]. FGF10 is involved in epicardial cell migration towards the compact myocardium. This FGF10 effect occurs paracrinally as myocardium-secreted FGF10 engages receptors on the epicardial cells (which later form cardiac fibroblasts). Thus, impairment in this FGF10 function indirectly elicits the declining proliferating potential of cardiomyocytes, the wall thickness in the right ventricle and heart size [16].

In a study on pluripotent stem cell experimental models, the interaction of FGF10-FGFR2 triggered the differentiation of cardiomyocytes [17]. However, the inducing proliferation capacity of FGF10 on cardiomyocytes in the intermediate developmental stage, using this exquisite pluripotent stem cell model, is largely unknown and is investigated in this study. In cardiomyocytes of adult mice (not in their cardiac fibroblasts), cell cycle re-entry is triggered by FGF10 [5]. Taken together, these studies have prompted us to underscore the different functional and structural effects of FGF10 in the commitment of cardiomyocytes from embryonic stem cells (ES).

On the other hand, CT-1, as a member of the interleukin-6 (IL-6) family, mediates pleiotropic roles through a heterodimer receptor of gp130/LIFRβ. CT-1 is expressed in the heart (including cardiac and non-cardiac cells), preserving the survival and proliferation of embryonic and neonatal cardiomyocytes. Commonly, CT-1 is known for its contribution to hypertrophy in adult cardiomyocytes, occurring as a compensatory response to hypertension. CT-1 secreted from non-cardiomyocytes exerts its function in cardiomyocytes by activation of the JAK-STAT signaling cascade and particularly STAT3 [18]. Moreover, according to our previous studies, CT-1 promotes the cardiomyogenesis of stem cells [19,20]. CT-1 modulates the expression of pro-cardiogenic growth factors, such as FGF-2, VEGF and PDGF-BB [20]. CT-1 and its receptor (gp130) contribute to the function of high molecular weight FGF-2 isoform on cardiomyocytes [21]. However, still little is identified about FGF ligand interactions with CT-1.

Cohen and co-workers related the modulatory function of FGF10 (as well as FGFs 3, 16 and 20) in the differentiation of myocardial progenitor cells to Wnt/β-catenin signaling [22]. Despite findings regarding activation of PI3K-AKT, p38 MAPK and FOXO3 associated with FGF10 function [5,23], little is still identified about involved downstream signaling elements of the FGF10 effect on cardiac cells. Nevertheless, in general, similar engaging signaling pathways (i.e., RAS-MAPK, PI3K-AKT and JAK-STAT) are involved in cardiac functions of FGFs and CT-1 [2,20]. Therefore, in the current study, we unraveled the inducing effect of FGF10 and CT-1 on cardiomyocyte proliferation from progenitor cells characterized by high expression of cardiomyocyte structural, transcriptional and proliferation markers. Our work opens horizons for the in-depth characterization of FGF10 in the propagation of cardiomyocytes applicable to regenerative therapies.

## 2. Materials and Methods

### 2.1. Materials

Recombinant mouse CT-1 was purchased from R&D Systems (Minneapolis, MN, USA). Recombinant murine FGF10 was obtained from PeproTech (Birmingham, Germany) and reconstituted in 5 mM sodium phosphate buffer (pH:7.4). Reconstitution buffer without FGF10/CT-1 was applied as the vehicle control.

### 2.2. Embryoid Body Formation and Contractile Activity Analysis during Cell Culture of ES Cells

Embryoid bodies were originated from ES cells (line CCE) and grown on mitotically inactivated feeder layers of primary murine embryonic fibroblasts to gain three-dimensional structures. Cells were cultured in Iscove’s medium (Gibco, Live Technologies, Helgerman Court, MD, USA) treated with 15% heat-inactivated fetal calf serum (FCS, 56 °C, 30 min) (AppliChem, Darmstadt, Germany), 100 μM of 2-mercaptoethanol (Sigma, Taufkirchen, Germany), 2 mM of glutamine (PAA Laboratories, Cölbe, Germany), 1% (*v*/*v*) NEA non-essential amino acids stock solution (100×) (Biochrom, Berlin, Germany), 1% (*v*/*v*) MEM amino acids (50×) (Biochrom), 1 mM of Na^+^-pyruvate (Biochrom), 2.5 μg/mL of plasmocin (InvivoGen, San Diego, CA, USA), 0.4% penicillin/streptomycin (100×) (Biochrom, Berlin, Germany), and 1000 U/mL of LIF (Chemicon, Hampshire, UK) at 37 °C in a humidified chamber with 5% CO_2_, and passaged every 2–3 days. Adherent cells were enzymatically dissociated using 0.05% trypsin-EDTA in phosphate-buffered saline (PBS) (Gibco, Helgerman Court, MD, USA), and seeded at a density of 3 × 10^6^ cells/mL in 250 mL siliconized spinner flasks (CellSpin, Integra Biosciences, Biebertal, Germany) containing 125 mL of the above mentioned supplemented Iscove’s medium (composition identical except plasmocin and LIF). To reach a 250 mL final volume, medium (125 mL) was added after 24 h. The spinner flask medium was stirred at 20 r.p.m. using a stirrer system (Integra Biosciences, Biebertal, Germany). From day 2 to day 4, 125 mL of the cell culture medium was exchanged daily with the same fresh medium. On day 4, approximately 30 embryoid bodies were plated into 6 cm cell culture dishes for each individual experiment. To evaluate the alterations in beating activities, the contraction frequencies of embryoid bodies were assessed within 1 min by microscopic inspection from day 9 to day 12. Maximum contractile activity of embryoid bodies was documented within 12 days of cell culture.

### 2.3. Immunocytochemistry and Confocal Imaging

Embryoid bodies were enzymatically dissociated with trypsin-EDTA (on day 9). Single cells were cultivated on coverslips. On day 12, cells were fixed in ice-cold methanol for 20 min at −20 °C, washed three times with PBS containing 0.01% Triton-X-100 (Sigma, Taufkirchen, Germany) (0.01% PBST) and permeabilized for 10 min with 1% PBST. Blocking against unspecific binding was carried out for 60 min with 10% heat-inactivated FCS dissolved in 0.01% PBST. The cells were subsequently incubated overnight with a monoclonal α-actinin antibody (Abcam, Cambridge, UK) (dilution 1:100) dissolved in blocking solution. Next, cells were washed three times with PBST (0.01% Triton) and reincubated for 1 h at room temperature in the dark with either Cy3- or Alexa 488-labeled anti-mouse antibodies (Abcam, Cambridge, UK) (dilution 1:100) in blocking solution. After washing in 0.01% PBST (three times), DRAQ5 (dilution 1:2000) (New England Biolabs, Frankfurt am Main, Germany) was applied for nuclear staining and excited at 633 nm. The cells were stored in PBS until inspection. Fluorescence images were acquired by means of a confocal laser scanning setup (Leica TCS SP2, Bensheim, Germany).

### 2.4. Quantitative Real-Time PCR- qPCR Analysis

Treated and control dissociated cells on day 12 were lysed in RLT Plus (Qiagen, Hilden, Germany). Next, RNA was extracted using the RNeasy Plus Micro kit (Qiagen), and cDNA synthesis was performed using the QuantiTect Reverse Transcription kit (Qiagen) according to the manufacturer’s protocol. After that, selected primers (Table 1) were designed via NCBI’s Primer-BLAST option (https://www.ncbi.nlm.nih.gov/tools/primer-blast/) (last accessed: 16 July 2020). Quantitative real-time polymerase chain reaction (qPCR) was performed using the PowerUp SYBR Green Master Mix kit (according to provided supplier instructions of Applied Biosystems) and the LightCycler 480 II machine (Roche Applied Science). Hypoxanthine-guanine phosphoribosyltransferase (Hprt) was utilized as a reference gene. Data were presented as mean expressions relative to Hprt. Data were assembled using GraphPad Prism software (GraphPad Software, La Jolla, San Diego, CA, USA).

### 2.5. Flow Cytometry

After treatment with FGF10 and CT-1, cells were dissociated using trypsin–EDTA, pelleted, fixed and permeabilized (eBioscience, Thermo-Fisher, 00-5523-00), resuspended in FACS buffer (0.1% sodium azide, 5% fetal calf serum (FCS), 0.05% in PBS), and stained with FITC-conjugated antibodies: α-actinin (sarcomeric) antibody- and its IgG-matched isotype control (both from Miltenyi Biotec, Bergisch Gladbach, Germany) for 30 min on ice in the dark, followed by washing. Flow cytometry data acquisition was carried out with the FACSCalibur™ instrument (BD) and BD CellQuest Pro software (BD Biosciences, San Jose, CA, USA). Data were analyzed using the FlowJo software, version X (FlowJo, LLC, Ashland, OR, USA). Data are represented as the percentage of positive-stained cells per total counted cells in each experiment.

### 2.6. Western Blot

After washing the isolated cells from embryoid bodies in PBS, the protein was extracted in RIPA lysis buffer (50 mM of Tris–HCl (pH 7.5), 150 mM of NaCl, 1 mM of EDTA (pH 8.0), 1 mM of glycerophosphate, 0.1% SDS, 1% Nonidet P-40) supplemented with protease inhibitor cocktail (Biovision, Hannover, Germany) and phosphatase inhibitor cocktail (Sigma) for 20 min on ice. After determination of the protein concentration using a Lowry protein assay, 20 μg of protein samples were boiled for 10 min at 70 °C, separated in NuPAGE 4–12% Bis–Tris gradient mini gels and transferred to nitrocellulose membranes by the XCell SureLock Mini-Cell Blot module (Thermo-Fisher, Frankfurt am Main, Germany) at 30 V and 180 mA for 90 min. Membranes were blocked with 5% (wt/vol) dry fat-free milk powder in Tris-buffered saline with 0.1% Tween (TBST) for 2 h at room temperature. Incubation with primary antibody was conducted at 4 °C overnight. The primary antibodies used were: monoclonal α-actinin (Sigma-Aldrich, Germany) and Troponin T2 (Abcam, UK), polyclonal MYL2 and MYL7 (Proteintech, Manchester, UK), monoclonal Aurora B and polyclonal GATA4 and NKX2-5 (Thermo-Fisher, Germany), monoclonal YAP-1 and vinculin (Proteintech, UK) and monoclonal β-actin (Cell Signaling Technology, Frankfurt am Main, Germany) antibodies. Subsequently, the membrane was incubated with horseradish peroxidase (HRP)-conjugated secondary antibodies (Abcam) for 60 min at room temperature. The blot was developed using a luminol-based ECL solution (consisting of 100 mM of Tris–HCl (pH 8.5), 900 μM of coumaric acid, 600 μM of luminol, and 880 μM of hydrogen peroxide). A generated chemiluminescence signal was quantified using the Peqlab gel documentation system (VWR International, Darmstadt, Germany). The density of the acquired protein bands on the Western blot image was assessed by ImageJ software (NIH, Bethesda, MD, USA). The final quantification represents the relative amounts of protein as a ratio of each target protein band to the corresponding housekeeping protein.

### 2.7. Statistical Analysis

Statistical analysis and graph assembly were performed using GraphPad Prism 9 (GraphPad Prism Software). The significance was determined by one-way ANOVA. Data are presented as mean ± SEM. Values of *p* < 0.05 were considered significant. The number of biological samples (n) for each group is stated in the corresponding figure legends.

## 3. Results

To investigate the putative effect of FGF10 and CT-1 on the proliferation capacity of cardiomyocytes, we treated ES for three consecutive days (day 9–11). This incubation time course aims to trigger cardiomyocyte maturation occurring in the intermediate developmental stage (IDS) (day 8–15 ± 2) of cardiomyogenesis. Based on our previous findings and others [17,20], cells were treated with 10 ng/mL (CT-1) and/or 100 ng/mL (FGF10), and consequently, alterations in the following cardiomyogenesis-related parameters were tested.

### 3.1. FGF 10 and CT-1 + FGF 10 Elevate Contraction Rate of Embryoid Bodies

The first insight on the cardiomyogenesis-activating role of FGF10 ± CT-1 was gained by analyzing the variations in beating rate per minute during and after incubation with the compounds. Of note, the FGF10, CT-1 and their combination enhanced spontaneous contraction rates compared to controls (Figure 1A). Statistical analysis on day 12 revealed a significant impact of FGF10 (109.87 ± 2.73) and CT-1 + FGF10 (106.75 ± 3.77) on beating elevation versus the vehicle (79.31 ± 1.76) and untreated control (79.12 ± 2.79) (Figure 1B). Seemingly, CT-1 demonstrated an increasing beating effect in this time course (98.75 ± 3.56) (Figure 1). FGF10 enhancing effect on contraction frequency of embryoid bodies was slightly higher than CT-1, but their differences were not statistically significant (Figure 1B).

### 3.2. FGF10 Increases the Proliferation Rate of Cardiomyocytes

The proliferating capacity of FGF10 on cardiomyocytes (d12) was evaluated on single cardiac cells dissociated from embryoid bodies (d9) treated with FGF10 from day 9 to day 11. After treatment, this enhancing effect was revealed by quantifying sarcomeric α-actinin, a common structural protein marker of cardiomyocyte-positive cells, to total cells detected by immunocytochemistry. Expectedly, the CT-1 treatment led to cardiomyogenesis (9.11 ± 1.58%) versus the vehicle control (4.31 ± 0.51%). Similarly, cardiomyocyte numbers significantly increased after FGF10 treatment (9.51 ± 0.72%). Intriguingly, co-administration of CT-1 and FGF 10 boosted the proliferation rate to substantially higher levels (14.72 ± 1.21%) (Figure 2A). This significant trend was supported by flow cytometry results, where a rise in α-actinin labeled cells after FGF10 (8.54 ± 0.50%) and CT-1 (6.75 ± 0.42%) versus the vehicle control (4.65 ± 0.14%) was observed. Flow cytometry data demonstrated a synergistic effect of FGF10 and CT-1 since a substantially greater number of positive cells (10.64 ± 0.39%) was detected after co-incubation of cells with FGF10 and CT-1 (Figure 2B), which is in agreement with the immunofluorescent image quantification results.

The alterations in the expression of sarcomeric α-actinin induced by FGF10/CT-1 were further confirmed in RNA and protein levels (Figure 3) which are to a great extent in line with immunocytochemistry and flow cytometry data (Figure 2). Substantial upregulation of α-actinin after treatment was recorded by qPCR analysis (Figure 3A). The Western blot data also showed an increasing impact of FGF10 + CT-1 on α-actinin levels (Figure 3B and Appendix A).

### 3.3. FGF10 and CT-1 Enhance Cardiomyocyte Structural, Transcriptional and Proliferation Gene Expression

The qPCR data demonstrate that FGF10 incubation, similar to CT-1, elevated the cardiomyocyte cell proliferation rate. This was observed through upregulation of the universal proliferation marker (Ki-67). In the context of cardiomyocyte proliferation, specific markers for cytokinesis (Aurora B) and transcriptional activation of proliferation (YAP-1) were assessed. The data of the present study showed that these two genes were highly expressed in treated cells compared with the controls. Although Ki-67 failed to demonstrate a substantial difference between single and combined FGF10 and CT-1 treatment, Aurora B and YAP-1 were substantially upregulated when cells were incubated simultaneously with both FGF10 and CT-1 (Figure 4A).

In agreement with the observed upregulation of α-actinin following FGF10 and CT-1 treatment (Figure 3), these incubations also significantly induced the gene expression of other cardiac-specific structural proteins (i.e., MLC-2a, MLC-2v, Troponin T2) (Figure 4B) and transcriptional factors (i.e., NKX2-5 and GATA4) (Figure 4C).

Despite weak detected signals in the Western blot of MLC-2v, Troponin T2, NKX2-5, and GATA4 proteins, a comparative analysis of the relative protein expression alterations following FGF10 and/or CT-1 incubation revealed an elevating trend of increasing expression following FGF10 and CT-1 treatment (Figure 5 and Appendix A). FGF10 particularly boosted GATA4 and NKX2-5 protein expression (Figure 5 and Appendix A). Intriguingly, a substantial elevation in structural protein levels (MLC-2a, MLC-2v and Troponin T2) under CT-1 + FGF10 incubation is in line with the upregulation of α-actinin (Figure 3 and Appendix A) and implies their synergistic contribution to cardiomyogenesis. This increasing trend occurred in proliferation markers (Aurora B and YAP-1); however, the differences in YAP-1 did not reach statistical significance (Figure 5 and Appendix A).

## 4. Discussion

In recent years, cardiac regenerative approaches have drawn research attention to employ FGFs for cardiac repair. Key strategies of cardiomyoplasty include injection or tissue-engineered implantation of progenitor stem cells, direct reprogramming of cardiac cells and induction of cell-cycle reactivation in adult cardiomyocytes [24]. In a previous study, Rubin et al. demonstrated that FGF10 increased numbers of epicardial cells but did not stimulate epithelial to mesenchymal transition. They also discussed that the proliferation-inducing effects of FGF10 were restricted on prenatal cardiomyocytes, whereas FGF10 expression and signaling effects are suppressed after birth [25]. In contrast, Rochais and colleagues demonstrated that mature cardiomyocytes in adult mice re-enter the cell cycle following overexpression of FGF10, suggesting a potential for regenerative therapy [5]. In support, previous findings revealed the FGF10 potential for induction of cardiomyocyte differentiation from stem cells [17]. The data of the present study underscore the impact of FGF10 on the proliferation of cardiomyocytes from pluripotent stem cells, which may open new horizons for cell replacement-based therapies.

Our data provided a body of evidence for cardiomyocyte proliferation after treatment with FGF10 and/or CT-1, which were based on increasing numbers of α-actinin- (a sarcomere marker) positive cells and the upregulation of cardiomyocyte-specific markers (Troponin T2, MLC-2v and MLC-2a, GATA4 and NKX2-5) (Figure 6). Criticism has been raised about the Ki-67 application as a cardiac proliferation marker due to being indistinguishable between increased polyploidization and cell division. Thus, apart from Ki-67, an assessment of markers of cytokinesis (e.g., Aurora B) and critical transcription factors in cardiomyocyte proliferation (e.g., YAP-1), together with α-actinin to detect cardiomyocyte proliferation and binucleation, were recommended [26]. Accordingly, we investigated the cardiomyocyte proliferation status by monitoring the differential expression of these indicators. Notably, all mentioned proliferation-related genes were upregulated following incubation with FGF10 and/or CT-1. Thus, in line with studies revealing that FGF10 functions prenatally and in vitro on the proliferation of progenitors and their differentiation to cardiomyocytes, the data of the current study support the notion that FGF10 promotes ES cell-derived cardiomyocyte proliferation, which may be applicable for cardiac regeneration.

The observed partial synergistic impact of co-treatment of CT-1 and FGF10 on cardiomyogenesis allows us to suggest a functional interaction between them. Previous studies demonstrated that FGF10, combined with FGF2 and vascular endothelial growth factor (VEGF), induced reprogramming of fibroblasts to cardiomyocyte-like cells by activating PI3-AKT and p38 MAPK signaling cascades [23]. As previously shown, cardiomyogenesis induced by exogenous CT-1 coincided with the elevation of intracellular second messengers, including calcium, ROS and NO. Importantly, endogenous CT-1 translocates to the nucleus, and this induced nuclear import has been suggested as a major characteristic of the CT-1 effect [20].

In the context of therapeutic applications, FGF10 and CT-1 integration in clinical interventions needs to be further investigated. Importantly, short stability and low bioavailability are major hindering hurdles in applying growth factors in clinical trials. Since the conjugation of FGF1 with biomaterials improved its stability after administration, this approach could be tested for an FGF10 drug delivery system to address availability and degradation problems [27,28]. Notably, CT-1 enhanced the delivery outcome of mesenchymal stem cell transplantation in myocardial infarcted models. This feature has been related to conferring focal adhesive characteristics to cells through modulation of the focal adhesion kinase [29,30]. Thus, this capacity of CT-1, together with the cardiomyogenic potential of FGF10 and CT-1, could be employed in designing stem cell therapies.

Depending on developmental stages, the modulatory effect of FGF10 on cardiomyocyte proliferation has been linked to interaction with FGFR2b in the embryonic developmental stage versus FGFR1b in the adult stage [4,5]. However, independent studies focusing on the interplay of involved FGFRs in cardiomyocyte response would shed light on the underlying mechanisms of FGF10 signal transduction. In addition to the identified region-specific proliferating effect of FGF10 on the right ventricle during development and regeneration in mouse models [4,5], future studies are still required to test the potential of FGF10 and CT-1 on other cardiac chambers during adult heart regeneration in humans.

## 5. Conclusions

Our experiments shed light on the inducing effects of FGF10 and CT-1 on the proliferation of cardiomyocytes originated from embryonic stem cells. We observed substantial increased cardiomyocyte numbers and upregulation in structural proteins, associated transcription factors, and cardiac cell proliferation markers following FGF10 and CT-1 treatment in the present study. The underlying signaling pathways involved in FGF10 and CT-1 functions on cardiomyogenesis, together with the application of the boosting potential of FGF10 and CT-1 on cardiomyocyte proliferation for cardiac regenerative therapies, need to be further studied.

## Figures and Tables

**Figure 1 biology-11-00534-f001:**
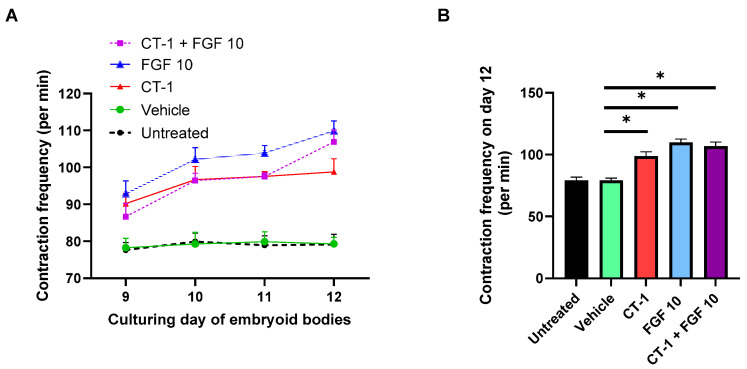
**Elevation in contraction rate after incubation with FGF10 and CT-1.** Embryoid bodies were treated from day 9 to day 11 with vehicle, FGF 10, CT-1 and CT-1+ FGF 10. (**A**) The beating frequency per min was measured from day 9 to day 12 in 4 independent experiments. (**B**) The contraction frequency differences among involved samples were compared on day 12. The data represent the means ± SEM. ^∗^
*p* < 0.05, significantly different from vehicle control (*n* = 4).

**Figure 2 biology-11-00534-f002:**
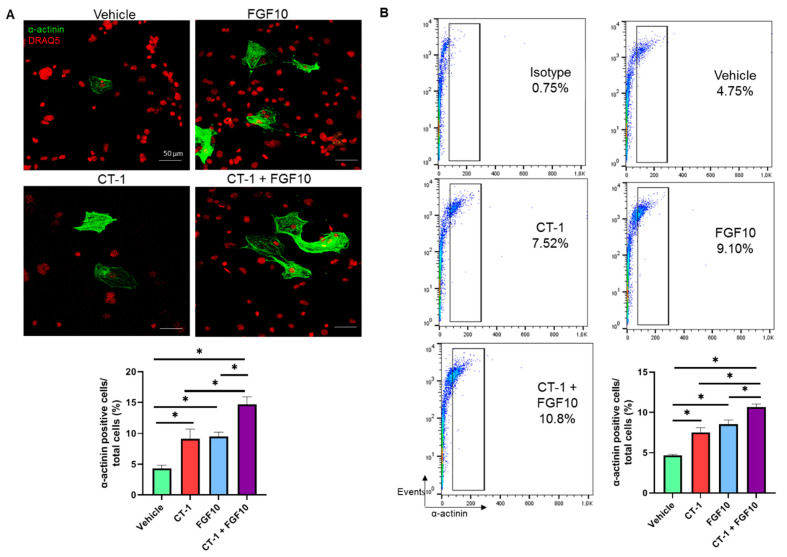
**Enhanced proliferation of cardiomyocytes after treatment with FGF10 and CT-1.** Embryoid bodies were enzymatically dissociated (day 9) and then treated with vehicle, FGF 10, CT-1 and FGF10 + CT-1 for 72 h. The number of sarcomeric α-actinin stained cells was analyzed by confocal microscopy and flow cytometry on day 12. (**A**) Dissociated cells were plated onto glass coverslips, treated for 3 days, fixed and fluorescently stained with α-actinin (green), and nuclei were counterstained using DRAQ5 (red). Subsequently, α-actinin positive cells were quantified. The scale bar represents 50 µm. (**B**) Representative flow cytometry of α-actinin- labeled cells. Data are presented as mean values ± SEM. ^∗^
*p* < 0.05, statistically significant as compared to vehicle control (*n* = 4).

**Figure 3 biology-11-00534-f003:**
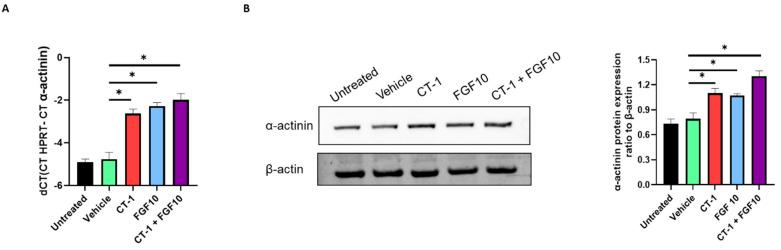
**Upregulation of α-actinin following treatment with FGF10 and CT-1.** Cells dissociated from embryoid bodies three days after treatment with FGF10 and/or CT-1, as well as vehicle, were subjected on day 12 to qPCR (**A**) and Western blot (**B**) for the analysis of α-actinin expression. (**B**) shows a representative Western blot. The bar charts demonstrate the quantification of independent experiments (*n* = 4). ^∗^
*p* < 0.05, significantly different from vehicle control (*n* = 4).

**Figure 4 biology-11-00534-f004:**
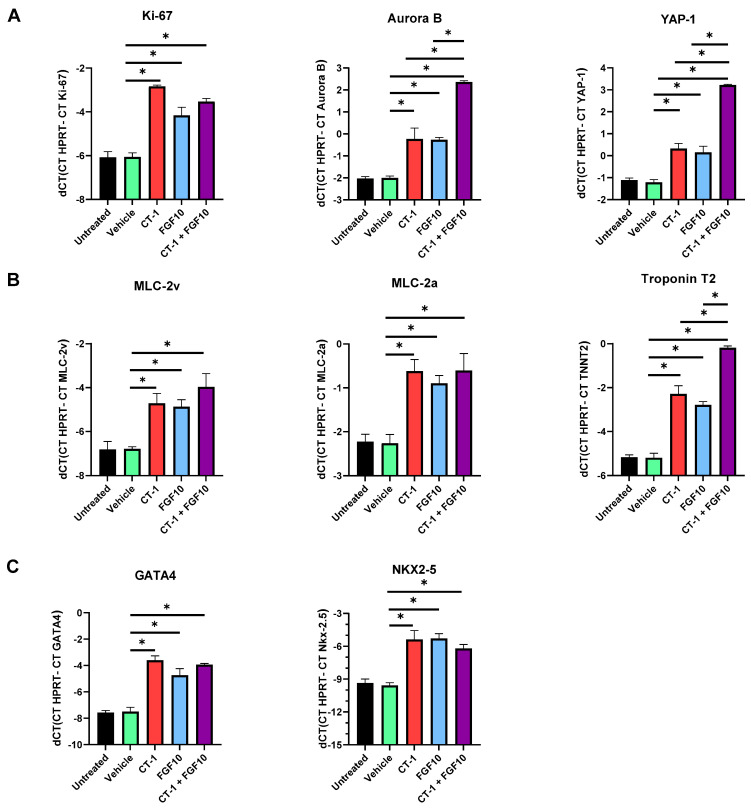
**Elevated mRNA expression of cardiomyocyte proliferation markers and cardiomyocyte-specific genes after treatment with FGF10 and CT-1**. Embryoid body-derived cells three days after incubation with FGF10 and/or CT-1, as well as vehicle, were subjected on day 12 to qPCR for the analysis of (**A**) proliferation markers (*Ki-67*, *YAP-1* and *Aurora B*), (**B**) cardiomyocyte structural genes (*MLC-2a*, *MLC-2v* and *Troponin T2* (*TTNt2*)), and (**C**) cardiomyocyte transcriptional factors genes (*GATA4* and *NKX2-5*). The bar charts demonstrate the quantification of independent experiments (*n* = 4). ^∗^
*p* < 0.05, significantly different from vehicle control.

**Figure 5 biology-11-00534-f005:**
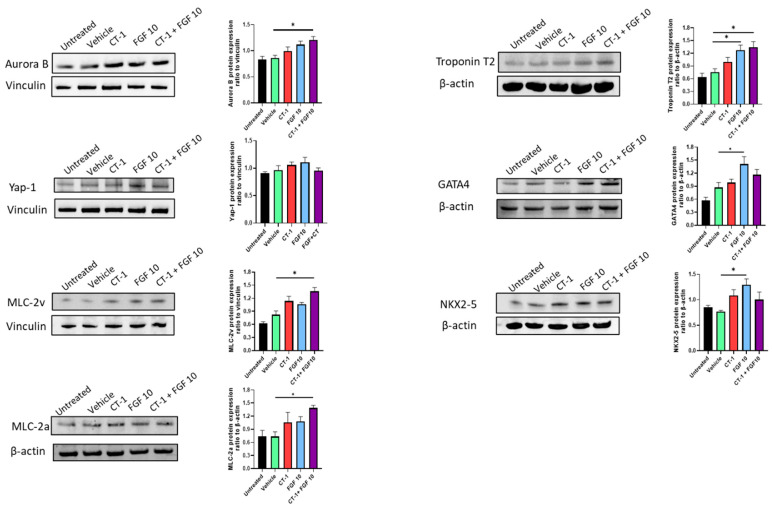
**Increased proteins levels of proliferation, structural and transcriptional markers in cardiomyocytes following FGF10 and CT-1 treatment.** Cells dissociated from embryoid bodies were treated from day 9 to day 12 with FGF10 and/or CT-1 as well as vehicle. Protein was extracted on day 12 and subjected to Western blot to analyze the status of Aurora B, YAP-1, MLC-2v, and MLC-2a, Troponin T2, GATA4 and NKX2-5. The panel demonstrates representative Western blots and their respective bar charts from quantification of independent Western blot experiments (*n* = 4). ^∗^
*p* < 0.05, significantly different from vehicle control.

**Figure 6 biology-11-00534-f006:**
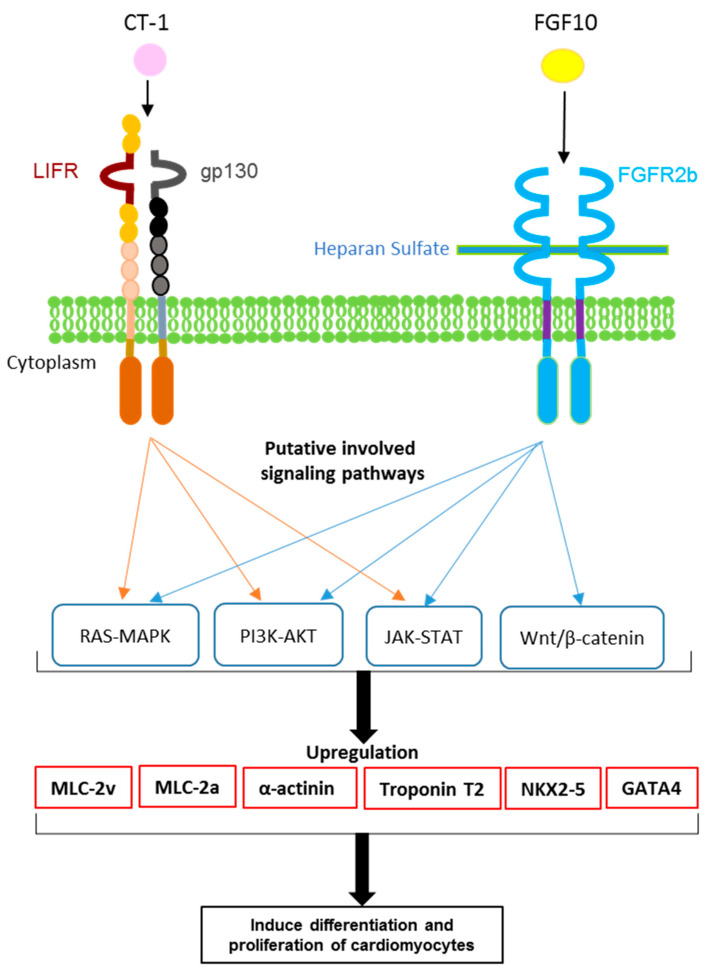
**Action mechanism of FGF10 and CT-1 on cardiomyocytes and putative involved underlying signaling pathways leading to the enhanced expression of cardiomyocyte-specific genes.** FGF10 and CT-1 mediate their functions through interaction with FGFR2b-heparan sulfate and LIFR/gp130, respectively. According to previous studies, RAS-MAPK [5,20,23], PI3K-AKT [5,20,23] and JAK-STAT [2,18,20] are common active pathways related to FGF10 and CT-1. FGF10 also contributes to the activation of Wnt/β-catenin [22]. We provided evidence that under FGF10 and CT-1 treatment, cardiomyocyte structural (α-actinin, Troponin T2, MLC-2v and MLC-2a) and transcriptional factors (NKX2-5 and GATA4) are being highly expressed, leading to induction of cardiomyogenesis.

**Table 1 biology-11-00534-t001:** Primer lists applied for qPCR analysis.

Gene	Forward Primer (5′->3′)	Reverse Primer (5′->3′)
Hprt	CCTAAGATGAGCGCAAGTTGAA	CCACAGGACTAGAACACCTGCTAA
Actn2	GTCAACACTCCCAAACCCGA	CTCCAACAGCTCACTCGCTA
Myl7	GGCACAACGTGGCTCTTCTA	GAACACTTACCCTCCCGA GC
MYL2	CTCCAAAGAGGAGATCGACCAG	TGTTTATTTGCGCACAGCCC
Yap-1	GAGCAAGCCATGACTCAGGA	CTCTGGTTCATGGCAAAACGA
Ki-67	CTGCGAGCTTCACCGAGAG	CAATACTCCTTCCAAACAGGCAG
Tnnt2	CCACATGCCTGCTTAAAGCTC	CTCGGCTCTCCCTCTGAAC
Aurkb	CGGGAGAAGAAGAGCCGTTT	AGGATGTTGGGATGTTTCAGGT
Gata4	GAGCAGGGGACAAGCCG	CGAAGCGGCAGTCCTGG
Nkx2-5	CCCAAGTGCTCTCCTGCTTT	AGCGCGCACAGCTCTTTT

## Data Availability

The data presented in this study are available in this article.

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
