# Peer review of "Induction of Stem-Cell-Derived Cardiomyogenesis by Fibroblast Growth Factor 10 (FGF10) and Its Interplay with Cardiotrophin-1 (CT-1)"

_biology, 2022, doi:10.3390/biology11040534_

Round 1

Reviewer 1 Report

In the study the effects of fibroblast growth factor 10 (FGF10) on induction of stem-cell-derived cardiomyogenesis and its interplay with cardiotrophin-1 were investigated.

There are several points which authors should explain and clarify. I have the following questions and comments:

1. You wrote (lines 234-236): “Statistical analysis on day 12 revealed a significant impact of FGF10 (109.87 ± 2.73 %) and CT-1 + FGF10 (106.75 ± 3.77 %) on beating elevation versus vehicle (79.31 ± 1.76 %) and untreated control (79.12 ± 2.79 %) (Figure. 1B).”

Are the values presented in figure really %? If yes, what is 100 %? Description of y-axis indicate that values represent beats per min (contraction frequency).

2. In Figures 3 and 5 are presented relative amounts of protein as a ratio of each target protein band to the corresponding housekeeping protein? Why did you use %?

3. In methods you mentioned that you used primary antibodies against the following proteins: α-actinin, Troponin T2, MYL2, MYL7, GATA4, vinculin, and β-actin. In Figure 5 are presented Western blot records also for other proteins: Aurora B, YAP-1, and NKX2-5.

4. The specificity of antibodies which were used in your study was in several cases not very good. See original Western blot records for MLC2v, Troponin T2, NKX2-5, and GATA4. From such records is sometimes not easy to come to relevant conclusion.

5. You did not isolate subcellular fractions (cytosolic, mitochondrial, nuclear) and for Western blot analysis you used the same cellular extract. From this point was interesting the use of different housekeeping proteins.  

6. Based on Figure 5 legend was quantification done from four independent experiments. In supplementary file with original blot images are presented blots from 1 experiment.

7. In Figure 6 are presented action mechanisms of FGF10 and CT-1. here is shown that Wnt/β-catenin pathway is related to FGF10 but not to CT-1. Do you have data about the changes in β-catenin protein levels in your experimental setting (FGF10 and/or CT-1 treatment)?

Author Response

We thank the reviewer for his/her evaluation and valuable comments.  We are grateful for the opportunity to revise this manuscript and believe that we could improve it by carefully considering your comments.

Point 1:  You wrote (lines 234-236): “Statistical analysis on day 12 revealed a significant impact of FGF10 (109.87 ± 2.73 %) and CT-1 + FGF10 (106.75 ± 3.77 %) on beating elevation versus vehicle (79.31 ± 1.76 %) and untreated control (79.12 ± 2.79 %) (Figure. 1B).” Are the values presented in figure really %? If yes, what is 100 %? Description of y-axis indicate that values represent beats per min (contraction frequency).

Response 1: We apologize for the oversight. As the reviewer correctly realized and we already demonstrated in Figure1, the beating frequency has been measured and represented in contraction per minute. The percentage symbol mistakenly had been remained from an older version of the manuscript, which we intended to show the statistics in percentage, but finally, we decided to demonstrate the contraction frequency alterations in contraction per minute. We have amended these mistakes by removing the percentage symbols (Page 5, Lines 236-239). To better clarify the unit of evaluations, we now mentioned "per minute "in the text (Page 5, Line 233).

Point 2:  In Figures 3 and 5 are presented relative amounts of protein as a ratio of each target protein band to the corresponding housekeeping protein? Why did you use %?

Response 2: In the previous publications of our lab (please see publications of Heinrich Sauer and colleagues ) we preferred representing the western blot quantifications in the percentage style in order to make it easier for readers to distinguish the differences between different test conditions. So we converted the relative ratios to percent. Now, raising this question from the reviewer prompted us to search the common and standard representation way of western blot data in the literature, which indeed outweigh demonstration graphs based on relative expression versus percentage, so our previous conversion to percent seems to be redundant. We thank the reviewer for this rhetorical question. As the reviewer implied, this point has now been addressed, and we represent the western blot quantification graphs in the relative ratio to the corresponding housekeeping protein (Figures 3 and 5 in pages 7 and 10).

Point 3: In methods you mentioned that you used primary antibodies against the following proteins: α-actinin, Troponin T2, MYL2, MYL7, GATA4, vinculin, and β-actin. In Figure 5 are presented Western blot records also for other proteins: Aurora B, YAP-1, and NKX2-5.

Response 3: We appreciate the reviewer for his/her care to find the missing information on the method section regarding Aurora B, YAP-1, and NKX2-5 antibodies. We now have provided the requested information (Page5, Lines 206-207).

Point 4: The specificity of antibodies which were used in your study was in several cases not very good. See original Western blot records for MLC2v, Troponin T2, NKX2-5, and GATA4. From such records is sometimes not easy to come to relevant conclusion.

Response 4: We understand the concern of the reviewer about the specificity of antibodies for the mentioned proteins, which could be in part due to the presence of unspecific bands on the membranes or weak recorded bands of interest. Despite purchasing brand new, claimed high-specificity antibodies and considering blocking and antibody titration measures, unfortunately, in some cases, these unspecific bands still remained, which, as the reviewer mentioned, raised concerns about the quality of antibodies. Please note that the unspecific bands are mainly distinct from the expected size for the protein of interest, and we validated the specificity of recorded bands. Secondly, the low protein expression and limited number of cardiomyocytes (approximately 5-15% of total cells) on day 12 could play a negative role in the weak appearance of target proteins on western blot membranes. To clarify this and address the opinion of the reviewer, we added now "Despite weak detected signals in western blot of MLC2v, Troponin T2, NKX2-5, and GATA4 proteins, a comparative analysis on the relative protein expression alterations following FGF10 and/or CT-1 incubation revealed" to the text (Page 9, Line 304-306).

Point 5: You did not isolate subcellular fractions (cytosolic, mitochondrial, nuclear) and for Western blot analysis you used the same cellular extract. From this point was interesting the use of different housekeeping proteins. 

Response 5: We are grateful for the advice and agree with the notion of considering different housekeeping proteins, especially when a housekeeping marker is not ubiquitously expressed among different samples. In fact, we used two housekeeping markers (beta-actin and vinculin) for all western blot membranes. As you see in Figure 5, our tested housekeeping proteins are ubiquitously expressed among different samples, and their size is distinct from the protein of interest. A technical limitation in using other housekeeping markers originates from the occurrence of undesired air bubbles on the western blot membranes in the areas where other housekeeping proteins are detectable, so we excluded considering the analysis of these housekeeping proteins with technical problems. We would like to clarify our consistency in the quantifications as we normalized the protein of interest concentration with the same housekeeping marker and membrane in all replicates for each target protein.

Point 6: Based on Figure 5 legend was quantification done from four independent experiments. In supplementary file with original blot images are presented blots from 1 experiment.

Response 6: Apparently, we misunderstood here. We thought the original representative images were meant to be uploaded in the supplementary file since we considered the fact that compiling all western blot images (56 images including all quantified proteins and corresponding housekeeping controls) in a single file could be large and exceed the defined capacity by the journal website. To address the request of the reviewer, we provide now in the attached file the original blots of all independent experiments in figure 5. We appreciate the reviewer's care about the accuracy of represented data in detail.

Point 7:  In Figure 6 are presented action mechanisms of FGF10 and CT-1. here is shown that Wnt/β-catenin pathway is related to FGF10 but not to CT-1. Do you have data about the changes in β-catenin protein levels in your experimental setting (FGF10 and/or CT-1 treatment)?

Response 7: As the reviewer 2 also raised this point, we agree that the scheme figure 6 was not precisely annotated since the signaling pathway-related references had not been mentioned, and this could lead to misunderstanding about distinguishing the main findings of this study versus general knowledge based on previous publications. In fact, we aimed to summarize different aspects of this paper, from background information to our results, in a comprehensive schematic figure applicable for both introduction (page 3, Lines 119 and 121) and discussion (Page 10, Line 339) sections. Indeed, we already had stated the respective references in the introduction (page 3, Lines 103-121) and discussion sections (Page 12, Line 365), and also discussed that independent studies are still required to investigate underlying signaling pathways (Page 12, Line 383-385). We now corrected this issue and included the respective references (which already were mentioned in the text) to Figure 6 and its legend as well as the terms of “Putative involved signaling pathways” and “according to previous studies” to Figure6 and its legend (Page 11, Lines 353-361) to discriminate clearly between our results and the information based on the literature. We thank the reviewers for emphasizing this important matter.

Reviewer 2 Report

In this manuscript, the authors demonstrated that FGF10 and/or CT-1 could induce cardiomyocyte proliferation in vitro. The mRNA and protein levels of several proliferation markers were evaluated. The potential mechanism was well discussed.  However, please see below for some comments:

  1. For the ES-derived CMs, the a-actinin was used for identifying the CM population. Since a-actinin (ACTN2) is expressed in both skeletal and cardiac muscles, it will be more convincing to use cardiac-specific markers such as cardiac troponin T to identify the CM population.

  1. In this manuscript, only the mRNA and protein level of proliferation markers was presented. With the additional support of immunofluorescence staining of PH3 and Aurora B, especially the symmetric Aurora B staining, it will make the conclusion more convincing.

  1. In figure 6, the authors presented the potential mechanism of how FGF10 and CT-1 could promote the proliferation of CMS, however, the activation of those signaling pathways was not directly demonstrated in this study.

Author Response

We thank the reviewer for his/her evaluation and valuable comments.  We are grateful for the opportunity to revise this manuscript and believe that we could improve it by carefully considering your comments.

Point 1:  For the ES-derived CMs, the a-actinin was used for identifying the CM population. Since a-actinin (ACTN2) is expressed in both skeletal and cardiac muscles, it will be more convincing to use cardiac-specific markers such as cardiac troponin T to identify the CM population.

Response 1: The reviewer is correct about the significance of cardiac troponin T in investigating cardiomyocyte population. Due to this rationale, we primarily considered analysis of cardiac troponin T alterations as one of the cardiomyocyte-specific structural genes (together with MLC-2a and MLC-2v) using qPCR and western blot (Figures 4 and 5). To our knowledge, sarcomeric α-actinin has also been widely applied in cardiomyocyte proliferation studies using embryonic stem cell models. We agree with the reviewer about the expression of α-actinin in skeletal muscle cells. However, we would like to explain the developmental and histological differences between these two cell types in our experimental model that convinced us about the specific presence of cardiomyocytes (but not skeletal muscle cells) for testing the α-actinin levels. During embryogenesis, the first organ system to be developed is the cardiovascular system. In embryoid bodies beating activity occurs already at day 7 of differentiation. In contrast, skeletal muscle differentiation takes place at later developmental stages, i.e. beyond the time window of cardiac cell differentiation investigated in the present study. Miller-Hance et al. (1993) have previously reported that skeletal muscle associated myogenin and myoD transcripts were evident during days 16-20 of ES cell differentiation, which demonstrates activation of the skeletal muscle program subsequent to the development of beating myocardium (Miller-Hance et al., 1993, JBC 268: 25244-25252). Moreover, skeletal muscle cells can be easily discriminated from cardiac cells in immunohistochemical experiments, since skeletal muscle cells form multinucleated myotubes from myoblasts, and the cell nuclei are located at the cell periphery. In contrast, the cardiac cells which were investigated in the present study showed only one or two nuclei which were located in the cell center.

Point 2:  In this manuscript, only the mRNA and protein level of proliferation markers was presented. With the additional support of immunofluorescence staining of PH3 and Aurora B, especially the symmetric Aurora B staining, it will make the conclusion more convincing.

Response 2: We appreciate the reviewer for this interesting suggestion. We followed the requested double staining with Aurora-B on the old samples (please find the attached figure in the attachment for reviewer 2). Despite the confirming trend of cardiomyocyte proliferation, the localization of Aurora-B did not seem to be highly specific. Thus, we are not convinced to use this figure in the paper. We are suspicious of the function of the Aurora-B antibody (Thermo Fisher) in this matter. Since the cultivation of the stem cells followed by an incubation period with FGF10 and CT-1 to test with another antibody from a different manufacturer is far beyond the defined deadline from the journal, due to this time limitation, unfortunately, we are not able to repeat this experiment.

Point 3: In figure 6, the authors presented the potential mechanism of how FGF10 and CT-1 could promote the proliferation of CMS, however, the activation of those signaling pathways was not directly demonstrated in this study.

As the reviewer 1 also raised this point, we agree that the scheme figure 6 was not precisely annotated since the signaling pathway-related references had not been mentioned, and this could lead to misunderstanding about distinguishing the main findings of this study versus general knowledge based on previous publications. In fact, we aimed to summarize different aspects of this paper, from background information to our results, in a comprehensive schematic figure applicable for both introduction (page 3, Lines 119 and 121) and discussion (Page 10, Line 339) sections. Indeed, we already had stated the respective references in the introduction (page 3, Lines 103-121) and discussion sections (Page 12, Line 365), and also discussed that independent studies are still required to investigate underlying signaling pathways (Page 12, Line 383-385). We now corrected this issue and included the respective references (which already were mentioned in the text) to Figure 6 and its legend as well as the terms of “Putative involved signaling pathways” and “according to previous studies” to Figure6 and its legend (Page 11, Lines 353-361) to discriminate clearly between our results and the information based on the literature. We thank the reviewers for emphasizing this important matter.

Round 2

Reviewer 1 Report

Authors gave explanation of points which were mentioned in my comments, and included several corresponding changes to the revised manuscript.

Reviewer 2 Report

The authors have addressed all of my concerns.